# COVID-19 preventive practices during intrapartum care- adherence and barriers in Ethiopia; a multicenter cross- sectional study

**Azmeraw Ambachew Kebede**[1]*, **Birhan Tsegaw Taye**[2], **Kindu Yinges Wondie**[1],
**Agumas Eskezia Tiguh**[1], **Getachew Azeze Eriku**[3], **Muhabaw Shumye Mihret**[1]

**1** Department of Clinical Midwifery, School of Midwifery, College of Medicine and Health Sciences, University of Gondar, Gondar, Ethiopia, **2** Department of Midwifery, College of Medicine and Health Sciences, Debre Berhan University, Debre Berhan, Ethiopia, **3** Department of Physiotherapy, College of Medicine and Health Sciences, University of Gondar, Gondar, Ethiopia

\* azmuzwagholic@gmail.com

## Abstract

### Background

Coronavirus disease 19 (COVID-19) is a newly emerging pandemic affecting more than 120 million people globally. Compliance with preventive practices is the single most effective method to overcome the disease. Although several studies have been conducted regarding COVID-19, data on healthcare provider's adherence to COVID-19 preventive practices during childbirth through direct observation is limited. Therefore, this study aimed to assess healthcare provider's adherence to COVID-19 preventive practices during childbirth in northwest Ethiopia.

### Methods

A multicenter study was conducted at hospitals in northwest Ethiopia among 406 healthcare providers from November 15 /2020 to March 10 /2021. A simple random sampling technique was employed to select the study subjects. Data were collected via face-to-face interviews and direct observation using a structured questionnaire and standardized checklist respectively. EPI INFO version 7.1.2 and SPSS 25 were used for data entry and analysis respectively. Binary logistic regression analyses were undertaken to identify associated factors. The level of significance was decided based on the adjusted odds ratio (AOR) with a 95% confidence interval at a p-value of $\leq 0.05$.

### Results

The proportion of healthcare providers having good adherence to COVID-19 preventive practices during intrapartum care was 46.1% (95% CI: 41.2, 50.9). Healthcare providers who had job satisfaction (AOR = 3.18; 95% CI: 1.64, 6.13), had smartphone and/or computer (AOR = 2.75; 95% CI: 1.62, 4.65), ever received training on infection prevention (AOR = 3.58; 95% CI: 2.20, 5.84), earned higher monthly income (AOR = 2.15; 95% CI: 1.30, 3.57), and worked at health facility in the urban area (AOR = 1.72; 95% CI: 1.07, 2.77) had a

**Data Availability Statement:** All relevant data for the finding is fully available without restriction

within the paper and its Supporting information files.

**Funding:** The authors received no specific funding for this work.

**Competing interests:** The authors have declared that no competing interests exist.

**Abbreviations:** AOR, Adjusted Odds Ratio; CI, Confidence Interval; COR, Crude Odds Ratio; COVID-19, Corona Virus Disease 19; IP, Infection Prevention; PPE, Personal Protective Equipment; SPSS, Statistical Package for Social Science; VIF, Variance Inflation Factor; WHO, World Health Organization.

significant association with adherence to COVID-19 preventive practices. Moreover, the most commonly mentioned barriers for not adhering to the preventive practice of COVID-19 were crowdedness of the delivery room, non-availability of personal protective equipment, and shortage of alcohol or sanitizer.

## Conclusion

The healthcare provider's adherence to COVID-19 preventive practices was low. Hence, stakeholders need to pay special attention to increase healthcare provides' job satisfaction. In addition, the provision of continuous training on infection prevention would be helpful. Furthermore, personal protective equipment, alcohol, and sanitizer supply must be provided for healthcare providers.

## Introduction

Coronavirus disease 19 (COVID-19) remains one of the global top public health threats. World Health Organization (WHO) declared that COVID-19 is a public health emergency following its emergence in Wuhan city, China [1]. Following that declaration, a number of COVID-19 preventive practices such as physical distancing, hand washing, alcohol and sanitizer use, closure of public places, and the use of face masks have been endorsed and applied worldwide [2–4]. Despite immense attention and efforts, a COVID-19 pandemic impedes devastating and catastrophic public impacts on all aspects disregard the populations' background [5, 6].

From December 2019 to February 16, 2021, there have been a total of more than 120 million confirmed cases of COVID-19 with greater than 2.6 million deaths due to the disease globally [7]. From the global confirmed cases, the Africa region accounts for more than 4 million confirmed cases of COVID-19 and greater than 100,000 deaths [7, 8]. The first case of COVID-19 in Ethiopia was reported on March 13, 2020 [9]. As of March 16, 2021, there were a total of 175,467 confirmed cases of COVID-19, 143,828 recovered cases and 2,550 of them lost their life due to the disease in Ethiopia [10]. However, clear data on the number of healthcare workers infected with COVID-19 is limited. Some data until September 2020 indicated that more than 570,000 healthcare providers were infected with coronavirus and more than 2500 were lost their lives due to the disease [6]. Countries these days are reporting more COVID-19 incidents in health facilities than previously in the pandemic [11].

So far, the increasing incidence of COVID-19 and the lack of effective treatment have put the community, especially the health professionals, in a state of panic [3, 12]. This is because healthcare professionals take the lion's share in preventing and treating the disease, and their risk increases as well [13]. The problem is supposed to be getting worse during childbirth due to crowdedness. Because the maternity ward is full of people including the laboring mothers, companions, and healthcare providers that make it difficult to properly follow the COVID-19 preventive strategies. Besides, labor is a stressful and terrifying condition for the mother, the family, and even the healthcare providers that might cause failure to comply with the COVID-19 prevention practices [14]. In this aspect, evidence recommends that every healthcare worker should adhere to the maternal care standards, such as women-centered and respectful maternity care, with appropriate COVID-19 prevention precautions including wearing extra personal protective equipment (PPE) and proper hand hygiene [15].

Data regarding the vertical transmission of COVID-19 from the mother to child during pregnancy is scarce. Pregnant women have reduced access to healthcare and health facility delivery since the onset of COVID-19 [16, 17]. A prospective cohort study in Nepal showed that the

impact of COVID-19 on maternal health services is variable; institutions with lower birth rates were performing well compared with institutions having a higher volume of birth [18]. Although pregnant mothers have an equal risk of getting the COVID-19 like anyone else, it is important to screen all laboring mothers and companions for COVID-19, reduce the number of visitors, and implement preventive practices during childbirth [19]. Besides, care must be taken to protect neonates from COVID-19 infection [20]. In this aspect, standard preventive precautions should be taken in the maternity ward by healthcare providers, laboring women, and companions [19]. Healthcare providers are strongly advised to use PPE during childbirth [19]. That means healthcare workers have a full right to getting PPE and a safe and less risky working environment to protect themselves and others from any infection related to their work [20]. Compared to other healthcare providers, providers working in the labor and delivery units have frequent and prolonged contact with patients. In this regard, empirical evidence indicated that spending a long time in a risky place increases the likelihood of acquiring the virus [21].

Coronavirus disease 19 impacts the welfare of individuals and existing data indicate that COVID- 19 is associated with fear, stress, burnout intention, and depression to health care providers [22–30]. Other studies reported that COVID-19 is correlated with decreased maternal healthcare access and poor perinatal outcomes [31, 32]. Less strong data showcase that COVID-19 is also associated with neonatal loss, low birth weight, and preterm birth [17, 33]. In Ethiopia, many studies have been conducted on healthcare provider's preventive practice towards COVID-19 [15, 34–38]. However, most of these studies were collected through a web-based study and there was a dearth of studies in which the data were collected through direct observation. Besides, there was a lack of evidence on health worker's compliance with COVID-19 preventive practices during childbirth. Therefore, this study aimed to assess healthcare provider's adherence to COVID-19 preventive practices during intrapartum care provision, associated factors, and barriers in Gondar province hospitals.

## Methods and materials

### Study design, area, and period

The present study was conducted at hospitals in Gondar province, northwest Ethiopia from November 15th /2020 to March 10th /2021. It was a multicenter observational cross-sectional study. The Gondar province includes South Gondar, Central Gondar, West Gondar, and North Gondar zones. There is an estimated 5, 137, 443 population in the province. Besides, there are a total of 22 hospitals in Gondar province (2 referral hospitals, 1 general hospital, and 19 primary hospitals). These hospitals are serving for about more than 10 million population in the four zones of Gondar province and surrounding zones such as the North Wollo zone, Waghimra zone, and some parts of the Tigray region.

### Study population and inclusion criteria

All healthcare providers working at the maternity ward in the selected health hospitals of Gondar province were the study population. These include medical doctors, midwives, and integrated emergency surgeon officers (IESO). Healthcare providers who were available at the time of data collection and who had a willingness to participate in the study were included.

### Sample size determination and sampling procedure

The sample size for this study was determined by using a single population proportion formula by considering the following assumptions: 95% level of confidence, 50% provider's adherence to COVID-19 preventive practices, and 5% margin of error.

$$n = \frac{(Z\alpha/2)^2 p(1-p)}{d^2} = n = \frac{(1.96)^{2*}0.5(1-0.5)}{(0.05)2} = 384.$$ Where, n = required sample size, α = level of significant, z = standard normal distribution curve value for 95% confidence level = 1.96, p = providers adherence to COVID-19 preventive practices during childbirth, d = margin of error. Taking a non-response rate of 10%, a total of 422 study participants was gained. Data were collected from 15 hospitals (2 tertiary hospitals, 1 general hospital, and 12 primary hospitals). The sampling frame was designed after getting the list of healthcare providers from each hospital. The total number of health professionals in the selected hospitals during the data collection period was 544. Then, the total sample size was distributed to each selected hospital proportionally. A simple random sampling technique was employed to choose the study subjects. Thus, the University of Gondar comprehensive specialized hospital (n = 93), Debre Tabor specialized hospital (n = 70), Debark General hospital (n = 33), Ambagiorgis primary hospital (n = 16), Dembia primary hospital (n = 20), Metema primary hospital (n = 30), Tach Giant primary hospital (n = 24), Lay Gaint primary hospital (n = 20), Gohala primary hospital (n = 16), Ebinat primary hospital (n = 10), Andabet primary hospital (n = 14), Delgi primary hospital (n = 10), Ayikel primary hospital (n = 16), Mekaneyesus primary hospital (n = 17), and Addis Zemen primary hospital (n = 17) were included in the study.

## Variables of the study

**Dependent variable.** Healthcare providers' adherence to COVID-19 preventive practices during intrapartum care (good/poor).

**Independent variables.** Socio-demographic variables: age, sex, marital status, having smartphones and/ or computers, and watching television (TV), reading a newspaper, and monthly income.

Workplace and profession-related variables: year of experience, professional category, intention to stay in the profession, job satisfaction, facility type, working time, type of facility, the workload in the delivery room, working part-time in private institutions, education while working, training on infection prevention (IP), and location of the health facility.

## Measurements and operational definitions

The dependent variable in this study was the healthcare provider's adherence to COVID-19 preventive practices. A total of 13 questions were prepared to assess the adherence level of healthcare providers towards COVID-19 preventive practices during intrapartum care: 1) did the healthcare provider wash his/her hand before examining the laboring women? 2) Did the healthcare provider use an alcohol based-hand rub before touching the laboring women? 3) Did the health care provider wear a face mask while caring for the women? 4) Did the health care provider wear protective eyewear/ splash guard while caring for the women? 5) Did the health care provider wear protective gowns or work uniforms while caring for the women? 6) Did the health care provider wear disposable gloves during examining the women? 7) Did the health care provider wear a disposable cape? 8) Did the health care provider limit number of visitors during labor and delivery? 9) Did the health care provider properly remove used/contaminated materials in the appropriate place (like facemask)? 10) Did the health care provider wash his/her hand after touching the woman? 11) Did the health care provider shakes the hands of any individual in the maternity ward? 12) Did the healthcare provider avoid touching his eye, nose, and mouth with the unwashed hand? 13) Did the healthcare provider keep in contact with others in the delivery ward? Each question has a "Yes" or "No" response giving a score of 13 and 0 which are the maximum and minimum scores respectively (i.e. a score of 1 was given for "Yes" except for question number 11, in which a score of 1 was given for "No

"and a score of 0 was given for the other "No"). Likewise, the healthcare provider's adherence to COVID-19 preventive practices was composited to good adherence (which was coded as "1" and poor adherence which was coded as "0"). Thus, based on the collective score calculated to measure healthcare provider's preventive practice, a score above the mean was considered as having good adherence [34].

**Job satisfaction.** Nine questions were prepared to assess the healthcare provider's job satisfaction level. Thus, healthcare providers who scored above the mean value were considered as satisfied, otherwise dissatisfied [39].

## Data collection tools, methods, and procedures

The data collection tool was prepared based on the WHO recommendation and reviewing the literature [2, 20, 37, 40] and modified to the local context and study objective. Afterward, a group of researchers assessed the questionnaire to evaluate its suitability. Then, the data was collected using a structured and pretested questionnaire and checklists through face-to-face interviews and direct observation, respectively. Efforts have been made to reduce the effect of observation on the healthcare provider's practice, i.e. the Hawthorne effect, by guaranteeing providers that data collection is anonymous and the performance of each healthcare provider could no longer be stated to their managers or shared in public. Healthcare providers were also advised on the purpose of the study, in which the study is aimed to improve COVID-19 preventive practices in the maternity ward, not to pass judgments on the practice observed during data collection. Besides, healthcare providers were not aware of what specific procedures and activities were on the checklists, so they could not prepare by any means. To minimize the effect of personal and professional relationships, the data collectors were from other areas, not from the same health facility. Moreover, healthcare providers were observed initially using the checklist and interviewed later on through a structured tool. The questionnaire comprises socio-demographic characteristics, professional and work-related characteristics, questions assessing the healthcare provider's compliance with COVID-19 preventive practices, and perceived barriers for non-adherence. In the meantime, supplies for COVID-19 prevention (mask and sanitizer) were provided for the data collectors and supervisors. Moreover, they were oriented to keep their distance at the time of the interview to prevent the transmission of COVID-19.

## Data quality control

Before the actual data collection, a pretest was done on 5% of healthcare providers outside of the study area. A total of 20 individuals have participated in the data collection and supervision process such as 15 Diploma and 5 BSc in midwifery holders collect and supervise the data, respectively. Training was given for 3 days about the overall data collection process and safety measures during the data collection. During the data collection, the questionnaire was checked for completeness by the supervisors.

## Data processing and analysis

The collected data were checked manually for completeness and consistency before data entry. Data entry was carried out using EPI INFO version 7.1.2 and analyzed using SPSS version 25. After that, the data were checked for errors, missing observation, and inconsistencies and managed accordingly. Frequency tables and graphs were used to present descriptive and analytic statistics. Numbers and percentages were used to describe categorical variables. The association between each explanatory variable and the dependent variable was checked by Pearson's chi-square test. The variance inflation factor (VIF) was used to test the

multicollinearity assumptions where VIF <10 was acceptable. Then, the binary logistic regression (both bivariable and multivariable) model was fitted to identify independent predictors, and variables having a p-value of less than 0.2 were included in the multivariable logistic regression analysis to handle the possible effect of confounders. In the multivariable logistic regression analysis, the AOR with its 95% CI and a p-value of $\leq 0.05$ was used to determine the significant association between the explanatory variables and the dependent variable.

### Ethical considerations

The study was conducted under the Ethiopian Health Research Ethics Guideline and the declaration of Helsinki. Ethical clearance was obtained from the Institutional Ethical Review Board (IRB) of the University of Gondar (**Reference number: V/P/RCS/05/413/2020**). A formal letter of administrative approval was gained from each selected hospital. Anonymous written informed consent was taken from each of the study participants after a clear explanation of the aim of the study. The study participants were also notified that they have the right to withdraw from the study at any time during the study.

## Results

### Socio-demographic characteristics

In this study, a total of 406 healthcare workers were involved, making a 96.4% response rate. The median age of the study participants was 27 years (minimum age 22 and maximum age 59). Of the study participants, 247/406 (60.8%) were within the age group of 26–30 years, 247/406 (67%) were male, and 210/406 (51.7%) had a professional experience of 3 to 5 years [Table 1].

### Workplace and profession-related characteristics

Among the participants, 214/406 (52.7%) were working from primary hospitals and 179 (44.1%) of them reported that the presence of workload in their hospital. About 167 (41.1%) of healthcare providers received training on IP [Table 2].

### Adherence to COVID-19 preventive practices during intrapartum

The overall adherence level of healthcare providers towards COVID-19 preventive practices during intrapartum care provision was 46.1% (95% CI: 41.2, 50.9). Nearly two-thirds of the healthcare providers properly remove contaminated or used materials in the appropriate places. About 261 (64.3%) healthcare providers limit the number of visitors in the delivery room. However, 50.2% of health workers shake the hands of somebody else in the maternity ward [Table 3].

### Barriers of COVID-19 preventive practices during intrapartum care

Healthcare providers were asked why they failed to adhere to the COVID-19 precautionary practices during labor and delivery. Accordingly, the most commonly reported reasons for the poor adherence to COVID–19 preventive practices were overcrowding of the delivery room (case overload), lack of facemask and other personal protective equipment (PPE), and non-availability of alcohol and sanitizer. About 109 of the study participants strongly believe that COVID-19 preventive practices are not effective [Fig 1].

**Table 1. Socio-demographic characteristics of study participants in hospitals of Gondar province, northwest Ethiopia, 2020/2021 (n = 406).**

| Characteristics | Category | Frequency | Percentage (%) |
|---|---|---|---|
| **Age of study participants** | ≤ 25 | 85 | 20.9 |
| | 26–30 | 247 | 60.8 |
| | ≥ 31 | 74 | 18.2 |
| **Sex of study participants** | Male | 272 | 67 |
| | Female | 134 | 33 |
| **Marital status** | Single | 164 | 40.4 |
| | Married | 242 | 59.6 |
| **Experience** | ≤ 2 | 140 | 34.5 |
| | 3–5 | 210 | 51.7 |
| | ≥ 6 | 56 | 13.8 |
| **Ever reading newspaper** | Yes | 207 | 51 |
| | No | 199 | 49 |
| **Ever watching TV** | Yes | 353 | 86.9 |
| | No | 53 | 13.1 |
| **Having smart phone or computer** | Yes | 256 | 63.1 |
| | No | 150 | 36.9 |
| **Educational level** | Midwifery diploma | 119 | 29.3 |
| | Midwifery degree | 243 | 59.8 |
| | Midwifery master's degree | 25 | 6.2 |
| | Others* | 19 | 4.7 |
| **Average monthly income** | < 5000 ETB | 140 | 34.4 |
| | 5001–10000 ETB | 239 | 58.9 |
| | >10001 ETB | 27 | 6.7 |

Note:

* General practitioners, Residents, and IESO.

## Factors associated with adherence to COVID-19 preventive practices

A multivariable logistic regression analysis depicts that healthcare providers who had job satisfaction, had a smartphone and/or computer, received training on infection prevention, earned higher monthly income, and worked at health facilities in the urban had a statistically significant association with adherence to COVID-19 preventive practices. Healthcare provider's adherence to COVID-19 preventive practices was 3.58 times higher among participants who received training on infection prevention as compared to their counterparts (AOR = 3.58, 95% CI: 2.20, 5.84). The odds of having good adherence to COVID-19 preventive practices among healthcare providers who had good job satisfaction was 3 times higher than those healthcare providers who had no job satisfaction (AOR = 3.18, 95% CI: 1.64, 6.13).

The likelihood of good adherence to COVID-19 preventive practices among healthcare providers who have a smartphone and/or computer was 2.75 times higher compared with their counterparts (AOR = 2.75; 95%CI: 1.62, 4.65). Likewise, healthcare works who have a monthly income of between 5001–1000 ETB were two times more likely to have had good adherence to COVID-19 preventive practices as compared to those providers who earned a monthly income of less than 5000 ETB (AOR = 2.15: 95% CI: 1.30, 3.57). Lastly, healthcare providers working at a health facility located in the urban area were 1.72 times more likely to adhere to COVID-19 preventive practices compared with the reference group (AOR = 1.72; 95% CI: 1.07, 2.77) [Table 4].

**Table 2. Workplace and profession related characteristics of study participants in hospitals of Gondar province, northwest Ethiopia, 2020/2021 (n = 406).**

| Characteristics | Category | Frequency | Percentage (%) |
|---|---|---|---|
| **Facility type** | Primary hospital | 214 | 52.7 |
| | General hospital | 32 | 7.9 |
| | Tertiary hospital | 160 | 39.4 |
| **Facility location** | Urban | 223 | 54.9 |
| | Semi-urban | 183 | 45.1 |
| **Job satisfaction** | Satisfied | 324 | 79.8 |
| | Unsatisfied | 82 | 20.2 |
| **Received infection prevention training** | Yes | 167 | 41.1 |
| | No | 239 | 58.9 |
| **Working time** | Day | 295 | 72.7 |
| | Night | 111 | 27.3 |
| **Having smart phone and/or computer** | Yes | 256 | 63.1 |
| | No | 150 | 36.9 |
| **Intention to stay in the profession** | Yes | 290 | 71.4 |
| | No | 116 | 28.6 |
| **Interest to work in the delivery room** | Yes | 340 | 83.7 |
| | No | 66 | 16.3 |
| **Workload load in the ward** | Yes | 179 | 44.1 |
| | No | 227 | 55.9 |
| **Working part-time at private health facility** | Yes | 51 | 12.6 |
| | No | 355 | 87.4 |
| **Education while working** | Yes | 151 | 37.2 |
| | No | 255 | 62.8 |

## Discussion

To the best of our knowledge, this is one of the few studies in Ethiopia that evaluate healthcare provider's adherence to COVID-19 preventive practices during childbirth through direct observation. Hence, the present study assessed healthcare provider's adherence to COVID-19 preventive practices, associated factors, and barriers during intrapartum care in northwest Ethiopia hospitals. More than half of the healthcare providers did not adhere to the recommended preventive practices of COVID-19. Job satisfaction, having training on infection prevention, having a higher monthly income, having a smartphone or/and computer, and working at a health facility located in the urban area were significant predictors of healthcare provider's compliance with COVID-19 preventive practices.

This study found that healthcare provider's adherence to COVID-19 preventive practices was 46.1%, which is comparable with a study conducted in Lebanon-49.7% [41]. The result is lower than findings from Nigeria-91.1% [42], China-79.44% [40], and Yemen-87.7% [13]. It is also lower as compared to studies conducted in Ethiopia including the Amhara region-62% [15], Addis Ababa-59.8% [43], Delgi primary hospital-59.4% [44], and Debretabor town-68.3% [45]. This discordancy might be due to differences in the study setting, and data collection techniques. Most of the aforementioned studies were collected data through a web-based and self-administered questionnaire. In this case, all the collected information was self-reported and might be filled carelessly even without performing the preventive practices. Despite the low cost and easy access of the study participants, a web-based study is highly prone to response bias [46]. Besides, we are not sure that whether the data is reached and filled in by the right person. However, our study was collected through direct observation using a

**Table 3. Healthcare provider's adherence to COVID-19 preventive practices during intrapartum care provision in hospitals of Gondar province, northwest Ethiopia, 2020/2021 (n = 406).**

| COVID-19 preventive practice questions | Yes (%) | No (%) |
|---|---|---|
| Did the health care provider wash his/her hand before examining the laboring women? | 239 (58.9) | 167 (41.1) |
| Did the health care provider use alcohol based hand rub before touching the laboring women? | 236 58.1%) | 170 (41.9) |
| Did the health care provider wear facemask while caring the women? | 247 (60.8) | 159 (39.2) |
| Did the health care provider wear protective eyewear/ splash guard while caring the women? | 175 (43.1) | 231 (56.9) |
| Did the health care provider wear protective gowns or work uniform while caring the women? | 222 (54.7) | (45.3) |
| Did the health care provider wear disposable glove during examining each woman? | 208 (51.2) | 198 (48.8) |
| Did the health care provider wear disposable surgical cap? | 185 (45.6) | 221(54.4) |
| Did the health care provider limit number of visitors during labor and delivery? | 261 (64.3) | 145 (35.7) |
| Did the health care provider properly remove used/contaminated materials in the appropriate place? | 266 (65.5) | 140 (34.5) |
| Did the health care provider wash his/her hand after touching the women? | 182 (44.8) | 224 (55.2) |
| Did the health care provider shakes hands of any individual in the maternity ward? | 204 (50.2) | 202 (49.8) |
| Did the healthcare provider avoid touching his eye, nose and mouth with unwashed hand? | 251 (61.8) | 155 (38.2) |
| Did the healthcare provider keep contact with others? | 229 (56.4) | 177 (43.6) |

checklist in a way that will decrease the chance of response bias. One of the main advantages of direct observation study is helping researchers to study a situation or an event as its natural state, thus helping us gain a better understanding of the condition [47]. The other explanation might be the current study was conducted a year after the emergence of COVID-19 wherein healthcare providers may adapt to the situation and ignore the preventive practices haphazardly.

The result of this study, however, was higher when contrasted to studies done somewhere else in Ethiopia including two studies in Addis Ababa-35.9% [37] and 33.3% [38], and northwest Ethiopia-38.73% [34]. The possible justification might be the time gap, data collection technique, the tool used to measure the outcome variable, and the study setting. The above studies collected the data 3 months after the occurrence of COVID-19 in Ethiopia; that means, most of the population may not have adequate information about the preventive methods and even may not believe the disease is too infectious. But the current study was conducted almost 12 months after the disease has occurred. This means that information regarding COVID-19 preventive practices has been given continuously that further helps healthcare providers to have good adherence to COVID-19 prevention practices in the health institutions. In addition, Ethiopian Telecommunication has a default mobile phone message during voice calls regarding COVID-19 preventive methods and the need to practicing those practices. The other explanation might be the aforesaid studies incorporated healthcare providers who were working both in the health center and hospitals, whereas the present study includes only hospital staff. Studies indicate that healthcare providers who are working in health centers are less knowledgeable as compared to providers who are working in hospitals [48]. The disparity

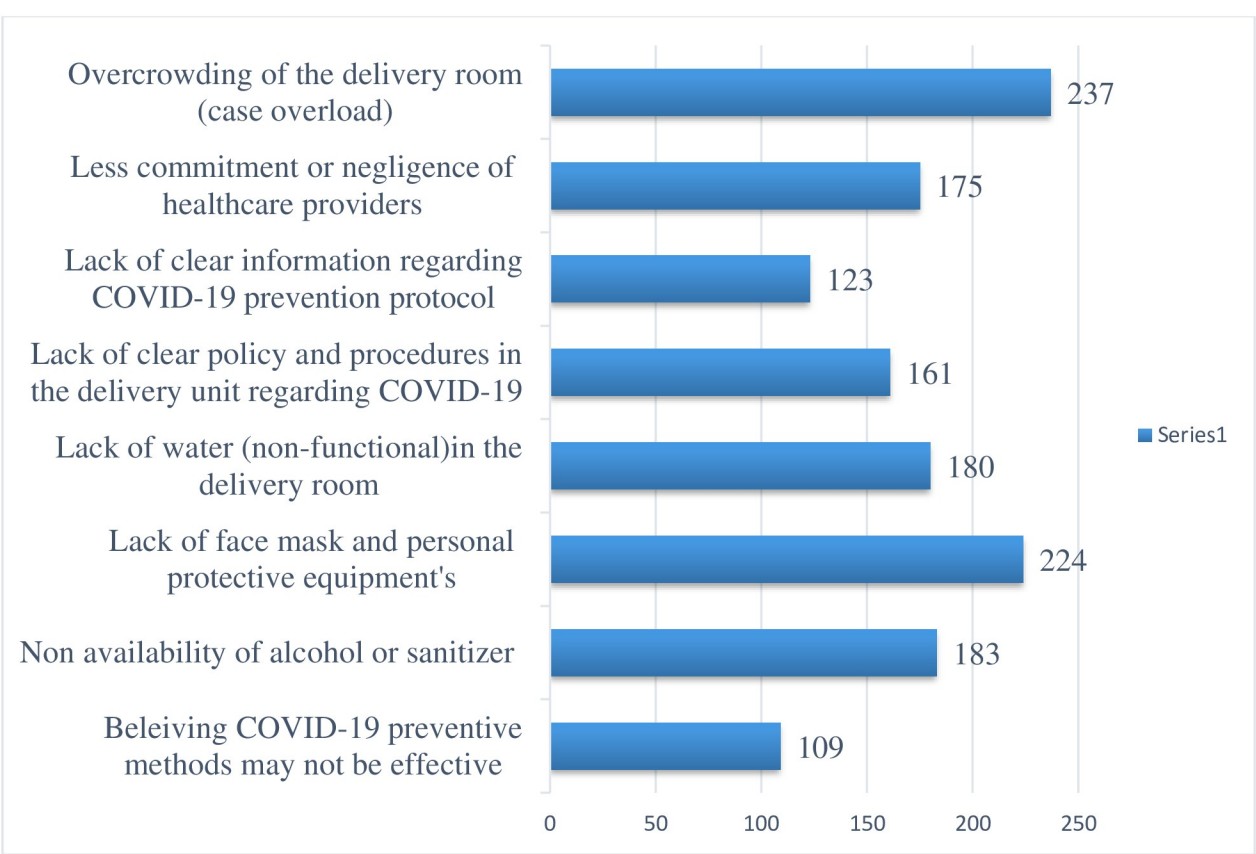

**Fig 1. Barriers for the poor adherence level of healthcare providers to the COVID-19 preventive measures during intrapartum care service provision in Gondar province Hospitals, northwest Ethiopia, 2021.**

might also be attributed to the tool used to measure the outcome variable. The two studies in Addis Ababa measured the outcome variable by using only alcohol-based hand rub and face mask application respectively. However, we measured the outcome variable using 13 questions which measures every aspect of COVID-19 prevention methods.

The current study indicated that having training on IP was a strong predictor of healthcare provider's adherence to COVI-19 preventive practices. Thus, healthcare providers who had ever received training on IP were 3.58 times more likely to have had good adherence to COVID-19 precautionary practices as compared to those who had no training. This finding is in agreement with a study conducted in Northwest Ethiopia, in which healthcare providers who were being trained with IP had good adherence to COVID-19 preventive practices [15]. Primarily, training on IP for healthcare providers is given to increase their knowledge regarding infection prevention, thereby maintaining a good level of infection prevention practice at the health institutions. Also, it is expected that healthcare workers having IP guiding principles at hand will have adequate knowledge on how to prevent and the consequences of failure to comply with IP strategies. Moreover, evidence highly recommends that consistent and exhaustive training is necessary for all healthcare workers to endorse readiness and crisis management like the COVID-19 pandemic [2, 49]. Concerning this, the federal and regional health bureau would better provide IP training for all healthcare providers promptly.

In this study, healthcare providers who had a smartphone and /or computer were 2.75 times more likely to have good adherence to COVID-19 preventive practices as compared to

**Table 4. Bi-variable and multivariable logistic regression analysis of factors associated with healthcare provider's adherence to COVID-19 prevention practices in hospitals of Gondar province, northwest Ethiopia, 2020/2021 (n = 406).**

| Variables | Category | COVID-19 prevention practice | | COR (95% CI) | AOR (95% CI) | p-values |
|---|---|---|---|---|---|---|
| | | Good | Poor | | | |
| Ever received infection prevention training | Yes | 113 | 54 | 4.66 (3.05, 7.13) | 3.58 (2.20, 5.84) | 0.000 |
| | No | 74 | 165 | 1 | 1 | |
| Watching TV | Yes | 172 | 181 | 2.41 (1.27, 4.53) | 0.50 (0.22, 1.13) | 0.079 |
| | No | 15 | 38 | 1 | 1 | |
| Read newspaper | Yes | 121 | 86 | 2.83 (1.89, 4.24) | 1.57 (0.95, 2.58) | 0.088 |
| | No | 66 | 133 | 1 | 1 | |
| Job satisfaction | Satisfied | 166 | 158 | 3.05 (1.77, 5.24) | 3.18 (1.64, 6.13) | 0.001 |
| | Unsatisfied | 21 | 61 | 1 | 1 | |
| Availability of internet | Yes | 119 | 113 | 1.64 (1.10, 2.44) | 0.76 (0.45, 1.27) | 0.305 |
| | No | 68 | 106 | 1 | 1 | |
| Monthly income | < 5000 ETB | 45 | 95 | 1 | 1 | |
| | 5001–10000 ETB | 133 | 106 | 2.64 (1.71, 4.10) | 2.15 (1.30, 3.57) | 0.003 |
| | > 10001 ETB | 9 | 18 | 1.05 (0.44, 4.51) | 0.40 (0.14, 1.08) | 0.071 |
| Professional category | Midwifery diploma | 43 | 76 | 1 | 1 | |
| | Midwifery degree | 124 | 119 | 1.84 (1.17, 2.89) | 1.12 (0.65, 1.93) | 0.683 |
| | Midwifery masters | 13 | 12 | 1.91 (0.80, 4.56) | 1.88 (0.50, 7.03) | 0.347 |
| | Others* | 7 | 12 | 1.03 (0.37, 2.81) | 0.69 (0.20, 2.31) | 0.554 |
| Having smart phone and/ or computer | Yes | 144 | 112 | 3.19 (2.07, 4.92) | 2.75 (1.62, 4.65) | 0.000 |
| | No | 43 | 107 | 1 | 1 | |
| Facility location | Urban | 124 | 99 | 2.38 (1.59, 3.57) | 1.72 (1.07, 2.77) | 0.024 |
| | Semi-urban | 63 | 120 | 1 | 1 | |

Notes:

* general practitioner, residents and integrated emergency surgeon officer.

Abbreviations: AOR, adjusted odds ratio; COR, crude odds ratio; CI, confidence interval; 1, reference category.

their counterparts. This finding is consistent with a study conducted in the Amhara region, Ethiopia [15]. This is because any information concerning COVID-19 has been released and updated through social media including Facebook, YouTube, Telegram, Email, Twitter, etc. Individuals having smartphones or/and computers can easily get new updates through social media using their smartphone or computer. Studies support that regular exposure to media increases people's conformity to COVID-19 preventive practices [50, 51].

Our study revealed that job satisfaction had a positive association with healthcare provider's adherence to COVID-19 prevention practices during childbirth. Hence, healthcare providers who had job satisfaction were three times more likely to have had a good COVID-19 preventive care as compared to their unsatisfied counterparts. Healthcare providers may get disappointed because of the working atmosphere, education opportunities, and professional allowance and benefits. If all these things are being fulfilled healthcare workers will do every activity properly and effectively. Studies support that job satisfaction enhances healthcare providers to adherence to the institution's values and missions [52].

Location of health facility was found to be significantly associated with healthcare provider's adherence to COVID-19 preventive methods. Accordingly, the odds of having a good adherence level towards COVID-19 cautionary practices during childbirth among healthcare providers working in health facilities located in the urban area was 1.72 times higher compared

with healthcare providers working in semi-urban areas. A similar finding was reported in a study conducted in northwest Ethiopia [15]. This could be explained by the accessibility of better training regarding infection prevention, and logistics for COVID-19 prevention (face mask, alcohol, sanitizer, and personal protective equipment) will be good in the urban area. In this study, more than half (224) of the study participants explained that one of the barriers to the non-adherence of healthcare providers to the COVID-19 preventive method was a lack of face masks and PPE. Moreover, most of the tertiary and general hospitals are located in the urban area and they are the center of referral. Thus, they are centers for surgery and neonatal intensive care units. All these situations need strict aseptic techniques and good infection prevention practices. This will further enhance health workers' commitment to comply with COVID-19 preventive practices.

Lastly, higher monthly income was found to be a strong predictor of healthcare provider's adherence to COVID-19 preventive methods. Consequently, the odds of having a good level of adherence towards COVID-19 preventive methods among healthcare providers who had a higher monthly income were two times higher as compared to their counterparts. Healthcare practitioners face many hazards including medico-legal issues, damage to medical instruments, and risk of acquiring infection, even death. An appropriate benefits packages and incentives for their work will encourage professionals to have a good level of adherence to any procedure in the health institution. Also, higher monthly income was found to be an enabling factor for good adherence to COVID-19 preventive practices so far [53]. Finally, we would like to point out that this study has some limitations. First, because of the cross-sectional nature of the study, it may be impossible to infer a causal-effect relationship between the outcome and the independent variables. Second, the study assessed a single time practice level that we could not confirm the true adherence of COIVD-19 preventive practices and may not be generalizable over time variations. The findings of this study, however, provide valuable information regarding healthcare provider's adherence to COVID-19 prevention practices despite the limitations.

## Implication for policymakers

The current study comes up with evidence on the context to which the COVID-19 preventive protocols during intrapartum care provision, barriers, and factors affecting compliance with COVID-19 preventive practices. Therefore, the findings from this study look for concerned stakeholders and health policymakers to pay special attention in executing the proper application of COVID-19 preventive practices during intrapartum care by ensuring healthcare providers multi-dimensional job satisfaction, appropriate salary, and befit packages for their work, providing continuous infection prevention training, and arranging educational opportunities so as to decrease staff burnout and intention to leave. The findings of this study also implicate for concerned bodies to prepare supplies necessary for COVID-19 prevention.

## Conclusion

The healthcare provider's adherence to COVID-19 preventive practices during intrapartum care was found to be low. It was positively predicted by job satisfaction, receiving IP training, earning better monthly income, having a smartphone and/or computer, and working at a health facility located in the urban area. Besides, the most commonly stated barriers for not adhering to the preventive practice of COVID-19 were case overload, non-availability of PPE, and scarcity of alcohol or sanitizer. The Ethiopian government and the ministry of health should emphasize healthcare provider's job satisfaction, appropriate payment for their work, and settle opportunities for regular media exposure and training. Moreover, personal

protective equipment, alcohol, and sanitizer supply must be provided for healthcare providers regularly.

## Supporting information

**S1 Questionnaire. English version.**
(DOCX)

**S1 Dataset. SPSS data.**
(SAV)

**S1 File. List of the full names of the selected health hospitals for this study.**
(DOCX)

## Acknowledgments

We would like to thank the University of Gondar for providing study ethical clearance to conduct this study. Our gratitude also goes to all data collectors and study participants. We are glad to Hospitals in Gondar province for writing permission letter.

## Author Contributions

**Conceptualization:** Azmeraw Ambachew Kebede, Muhabaw Shumye Mihret.

**Data curation:** Azmeraw Ambachew Kebede, Birhan Tsegaw Taye, Kindu Yinges Wondie, Agumas Eskezia Tiguh, Getachew Azeze Eriku, Muhabaw Shumye Mihret.

**Formal analysis:** Azmeraw Ambachew Kebede, Birhan Tsegaw Taye, Kindu Yinges Wondie, Agumas Eskezia Tiguh, Getachew Azeze Eriku, Muhabaw Shumye Mihret.

**Investigation:** Azmeraw Ambachew Kebede, Birhan Tsegaw Taye.

**Methodology:** Azmeraw Ambachew Kebede, Birhan Tsegaw Taye, Kindu Yinges Wondie, Getachew Azeze Eriku, Muhabaw Shumye Mihret.

**Validation:** Azmeraw Ambachew Kebede, Birhan Tsegaw Taye, Agumas Eskezia Tiguh, Getachew Azeze Eriku, Muhabaw Shumye Mihret.

**Visualization:** Azmeraw Ambachew Kebede, Birhan Tsegaw Taye, Kindu Yinges Wondie, Agumas Eskezia Tiguh, Getachew Azeze Eriku, Muhabaw Shumye Mihret.

**Writing – original draft:** Azmeraw Ambachew Kebede.

**Writing – review & editing:** Azmeraw Ambachew Kebede, Birhan Tsegaw Taye, Agumas Eskezia Tiguh, Getachew Azeze Eriku, Muhabaw Shumye Mihret.

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
