## [Decision Letter · Decision Letter 0]

18 Jun 2021

PONE-D-21-09407

COVID-19 preventive practices during intrapartum care; adherence, associated factors and barriers in the case of northwest Ethiopian healthcare providers: a multicenter study

PLOS ONE

Dear Dr. Kebede,

Thank you for submitting your manuscript to PLOS ONE. After careful consideration, we feel that it has merit but does not fully meet PLOS ONE’s publication criteria as it currently stands. Therefore, we invite you to submit a revised version of the manuscript that addresses the points raised during the review process.

ACADEMIC EDITOR: The academic editor served as the second reviewer on this manuscript and agree that major revisions, especially methods, needs to be addressed for further publication consideration.

We look forward to receiving your revised manuscript.

Kind regards,

Joseph Telfair, DrPH, MSW, MPH

Academic Editor

PLOS ONE

Journal Requirements:

2. Please include your actual numerical p-values in Table 4.

3. Please provide a list of the full names of the selected health hospitals for this study (you may provide this as a supplementary file).

Additional Editor Comments (if provided):

The academic editor served as the second reviewer on this manuscript and agree that major revisions, especially methods, needs to be addressed for further publication consideration.

Reviewers' comments:

Reviewer's Responses to Questions

**Comments to the Author**

1. Is the manuscript technically sound, and do the data support the conclusions?

Reviewer #1: Yes

2. Has the statistical analysis been performed appropriately and rigorously? 

Reviewer #1: No

3. Have the authors made all data underlying the findings in their manuscript fully available?

Reviewer #1: No

4. Is the manuscript presented in an intelligible fashion and written in standard English?

Reviewer #1: Yes

5. Review Comments to the Author

Reviewer #1: Thank you for inviting me to review this research. I read and review the research with great interest. The research posed on infection prevention practice during pandemic is important. Researchers has aimed to explore factor associated with infection prevention practice. I have four major comments mainly 1) objective, 2) design, 3) data collection process and data analysis.

1. Objective- It is important to assess whether COVID-19 pandemic prompted improvement or change in infection practice during labour and childbirth. This can be done through observation before and during COVID-19 pandemic. I see that the study collected data through observation on infection prevention practice during pandemic. This is one of the limitation of the study. Can the researcher collect data on maternal and neonatal infection rate in these hospitals before and during pandemic.

2. Design- Can researcher make it a retrospective cohort study, with change in infection rate before and during pandemic,

3. Data collection- One key variable of interest is did the infection prevention practice change by obstetric characteristics (mode of birth, complication during labour). Can the data collection be made on mode of birth. Further, the data collection on the continuous supply of PPE is important, can research provide the supply chain system.

4. Data analysis- Can the inter-hospital infection prevention practice variance be provided.

5. So, what the the policy implication of the study.

6. There are typos in the research, if accepted needs a thorough english language review.

6. PLOS authors have the option to publish the peer review history of their article (what does this mean?). If published, this will include your full peer review and any attached files.

Reviewer #1: No

---

## [Decision Letter · Decision Letter 1]

31 Aug 2021

PONE-D-21-09407R1

COVID-19 preventive practices during intrapartum care; adherence, associated factors and barriers in the case of northwest Ethiopian healthcare providers: a multicenter study

PLOS ONE

Dear Dr. Kebede,

Thank you for submitting your manuscript to PLOS ONE. After careful consideration, we feel that it has merit but does not fully meet PLOS ONE’s publication criteria as it currently stands. Therefore, we invite you to submit a revised version of the manuscript that addresses the points raised during the review process.

ACADEMIC EDITOR:  

The academic editor has served as the second reviewer for this manuscript and agree minor revision are warranted. 

We look forward to receiving your revised manuscript.

Kind regards,

Joseph Telfair, DrPH, MSW, MPH

Academic Editor

PLOS ONE

Journal Requirements:

Reviewers' comments:

Reviewer's Responses to Questions

**Comments to the Author**

1. If the authors have adequately addressed your comments raised in a previous round of review and you feel that this manuscript is now acceptable for publication, you may indicate that here to bypass the “Comments to the Author” section, enter your conflict of interest statement in the “Confidential to Editor” section, and submit your "Accept" recommendation.

Reviewer #1: All comments have been addressed

2. Is the manuscript technically sound, and do the data support the conclusions?

Reviewer #1: Yes

3. Has the statistical analysis been performed appropriately and rigorously? 

Reviewer #1: Yes

4. Have the authors made all data underlying the findings in their manuscript fully available?

Reviewer #1: No

5. Is the manuscript presented in an intelligible fashion and written in standard English?

Reviewer #1: Yes

6. Review Comments to the Author

Reviewer #1: Thank you for inviting to re-review the research work. The work has substantially improved. Here are my minor suggestion for improvement.

1. The title needs to be shortened can it be "COVID-19 preventive practices during intrapartum care-adherence and barriers in Ethopia; a multi-centered cross-sectional study."

2. In the abstract, findings please remove word "statistically" as significant association does imply that. Please add the reference on "Perfect storm by KC A et al" http://www.jogh.org/documents/2021/jogh-11-05010.htm which shows improvement in infection prevention practices.

7. PLOS authors have the option to publish the peer review history of their article (what does this mean?). If published, this will include your full peer review and any attached files.

Reviewer #1: No

---

## [Decision Letter · Decision Letter 2]

5 Oct 2021

PONE-D-21-09407R2COVID-19 preventive practices during intrapartum care- adherence and barriers in Ethiopia; a multicenter cross-sectional studyPLOS ONE

Dear Dr. Kebede,

Thank you for submitting your manuscript to PLOS ONE. After careful consideration, we feel that it has merit but does not fully meet PLOS ONE’s publication criteria as it currently stands. Therefore, we invite you to submit a revised version of the manuscript that addresses the points raised during the review process.

ACADEMIC EDITOR:   The academic editor served as the second reviewer for the manuscript and agree the minor issues must be address in order to move forward with consideration for publication. 

We look forward to receiving your revised manuscript.

Kind regards,

Joseph Telfair, DrPH, MSW, MPH

Academic Editor

PLOS ONE

Journal Requirements:

Reviewers' comments:

Reviewer's Responses to Questions

**Comments to the Author**

1. If the authors have adequately addressed your comments raised in a previous round of review and you feel that this manuscript is now acceptable for publication, you may indicate that here to bypass the “Comments to the Author” section, enter your conflict of interest statement in the “Confidential to Editor” section, and submit your "Accept" recommendation.

Reviewer #1: All comments have been addressed

2. Is the manuscript technically sound, and do the data support the conclusions?

Reviewer #1: Partly

3. Has the statistical analysis been performed appropriately and rigorously? 

Reviewer #1: No

4. Have the authors made all data underlying the findings in their manuscript fully available?

Reviewer #1: Yes

5. Is the manuscript presented in an intelligible fashion and written in standard English?

Reviewer #1: Yes

6. Review Comments to the Author

Reviewer #1: Thank you for inviting me again to review this interesting and improved revised work.

As I revisited the manuscript and result, I noted that health care provider who earn better has been found to be associated with better care. This is a confounder, since health worker who earn more might be better professionally educated such as specialist doctor. So, I suggest to assess the association between education and control earning variable. I suggest to conduct the association between training in table 2 and conduct either multi variate or multi level association with care i.e in table 4.

The reference in 53 to explain why health workers earning more with better intrapartum care, just not well explain the association.

Further, the methodological consideration is not provided, please kindly provide the limitation of the study especially

the design.

7. PLOS authors have the option to publish the peer review history of their article (what does this mean?). If published, this will include your full peer review and any attached files.

Reviewer #1: **Yes: **Ashish KC

---

## [Decision Letter · Decision Letter 3]

8 Nov 2021

COVID-19 preventive practices during intrapartum care- adherence and barriers in Ethiopia; a multicenter cross-sectional study

PONE-D-21-09407R3

Dear Dr. Kebede,

We’re pleased to inform you that your manuscript has been judged scientifically suitable for publication and will be formally accepted for publication once it meets all outstanding technical requirements.

Kind regards,

Joseph Telfair, DrPH, MSW, MPH

Academic Editor

PLOS ONE

Additional Editor Comments (optional):

The academic editor served as the second reviewer and agree to accept the manuscript.

Reviewers' comments:

Reviewer's Responses to Questions

**Comments to the Author**

1. If the authors have adequately addressed your comments raised in a previous round of review and you feel that this manuscript is now acceptable for publication, you may indicate that here to bypass the “Comments to the Author” section, enter your conflict of interest statement in the “Confidential to Editor” section, and submit your "Accept" recommendation.

Reviewer #1: All comments have been addressed

2. Is the manuscript technically sound, and do the data support the conclusions?

Reviewer #1: Yes

3. Has the statistical analysis been performed appropriately and rigorously? 

Reviewer #1: Yes

4. Have the authors made all data underlying the findings in their manuscript fully available?

Reviewer #1: Yes

5. Is the manuscript presented in an intelligible fashion and written in standard English?

Reviewer #1: Yes

6. Review Comments to the Author

Reviewer #1: Thank you for the commitment to improve your research work. That manuscript has improved and have addressed all of my comments.

Best wishes

7. PLOS authors have the option to publish the peer review history of their article (what does this mean?). If published, this will include your full peer review and any attached files.

Reviewer #1: **Yes: **Ashish KC

---

## [Editor Report · Acceptance letter]

10 Nov 2021

PONE-D-21-09407R3 

COVID-19 preventive practices during intrapartum care- adherence and barriers in Ethiopia; a multicenter cross- sectional study 

Dear Dr. Kebede:

I'm pleased to inform you that your manuscript has been deemed suitable for publication in PLOS ONE. Congratulations! Your manuscript is now with our production department. 

Kind regards, 

on behalf of

Dr. Joseph Telfair 

Academic Editor

PLOS ONE